# Fast Assembly of Metal Organic Framework UiO-66 in Acid-Base Tunable Deep Eutectic Solvent for the Acetalization of Benzaldehyde and Methanol

**DOI:** 10.3390/molecules27217246

**Published:** 2022-10-25

**Authors:** Lifang Chen, Xiangzhu Ye, Ting Zhang, Hao Qin, Hongye Cheng, Zhiwen Qi

**Affiliations:** State Key Laboratory of Chemical Engineering, School of Chemical Engineering, East China University of Science and Technology, Meilong Road 130, Shanghai 200237, China

**Keywords:** deep eutectic solvent, metal organic framework, UiO-66, acid-base, acetalization

## Abstract

Zirconium-based metal-organic frameworks (MOFs) have attracted extensive attention owing to their robust stability and facile functionalization. However, they are generally prepared in common volatile solvents within a long reaction time. Here, we introduced environmentally friendly, cheap, and acid-based tunable deep eutectic solvents (DESs) formed from 2-methyl imidazole (MIm) and p-toluenesulfonic acid (PTSA) which significantly accelerated the assembly of zirconium-based MOF (UiO-66) without any aggressive additives. PTSA in acidic DES and ZrOCl_2_ preliminarily formed Zr(IV) oxo organic acid framework, whereas basic DES completely dissolved the ligand of UiO-66. The strong hydrogen bond effect of PTSA and MIm efficiently accelerated the linker exchange between zirconium oxo organic coordination in acidic DES and benzenedicarboxylate linker in weak basic DES within a reaction time of 2 h at 50 °C. Thus, UiO-66 was quickly assembled with small particle sizes and used as an excellent catalyst for the acetalization of benzaldehyde and methanol. Therefore, the developed synthesis approach provides a new green strategy to quickly prepare and design various structures of metal-based compounds under mild reaction conditions.

## 1. Introduction

Metal organic frameworks (MOFs) with organic inorganic hybrid compositions, high surface areas, and versatile porous structures have gained extensive attention in gas adsorption and storage [1,2], separation [3], drug delivery [4,5], and catalysis [6,7,8]. Particularly, they possess steric characters of active metal sites and adjustable pore sizes, facilitating accessibility and transport of catalytic substrates and products [9,10,11,12]. With the advantages of homogeneous and heterogeneous catalysts, they exhibit high activity, selectivity, and stability for a series of catalytic processes. Among them, zirconium-based MOFs have the potential for applications owing to their excellent stability and facile functionalization [13]. Generally, they are synthesized through conventional solvothermal methods with a long reaction time and high reaction temperature [14,15]. Extensive efforts have been focused on accelerating their formation, such as the addition of aggressive additives [16,17], microwave [18], and mechanochemical milling [19], which are energy intensive and complex due to product contamination and impurity removal. In contrast, environmentally benign approaches with low energy and much shorter times to prepare Zr-based MOFs are desirable.

Ionic liquids (ILs), as green solvents, are widely used in the synthesis of metal oxides, and MOFs due to their unique properties [20,21,22,23]. They have many advantages, such as acting as both solvent and structure direction agent with tunable hydrophilic and hydrophobic property, dissolving organic and inorganic precursors, thermal stability, environmental benignity, and so on [24]. For example, Zhang et. al. found the second component can promote the formation of MOFs in ILs [25,26]. Recently, they have also reported that aprotic IL 1-hexyl-3-methylimidazolium chloride as solvent could significantly accelerate the room-temperature formation of Zr-based MOFs with the help of acetic acid within only a half hour [27]. We also previously prepared manganese-based oxides catalysts under the assistance of ILs with efficient catalytic activity for selective oxidation of 5-hydroxymethylfurfural [20,21]. However, ILs with complex preparation procedures, low biodegradability, and high cost have limited their practical applications.

Accordingly, as environmentally friendly, cheap, and acid-based tunable solvents, deep eutectic solvents (DESs) with the advantages of ILs are continuously emerging as analogous ILs [28,29,30,31]. Composed of a hydrogen bond acceptor (HBA, e.g., quaternary ammonium or phosphonium salt) and a hydrogen bond donor (HBD, e.g., organic acid or alcohol), DESs can greatly decrease freezing points relative to individual components due to the formation of hydrogen bonds between the resultants. They have been reported as reaction media for many inorganic metals, metal oxides, metal chalcogenides, metal phosphate, organic carbonides and nitrides [32,33,34]. However, very few examples of MOFs prepared from DESs using general synthetic approach have been reported, and only zeolitic imidazolate framework using DES formed from 2-methyl imidazole (MIm) and tetrabutylammonium bromide has been proposed [35]. Owing to their environmental benignancy, facile preparation, and low cost, it is attractive to utilize DESs as solvents to prepare MOFs under mild conditions.

Based on this motivation, for the first time, we contribute a green MOF preparation strategy in acid-based tunable DES environment formed from MIm and p-toluene sulfonic acid (PTSA). The tuned acidic and basic DESs can significantly accelerate the assembly of zirconium-based metal–organic framework (University of Oslo, UiO-66) with high yield under mild conditions without any aggressive additives. The MOF demonstrated high catalytic activity for the acetalization of benzaldehyde and methanol.

## 2. Material and Methods

### 2.1. Chemicals

Hydrated zirconyl nitrate (ZrOCl_2_·8H_2_O), N,N-dimethylformamide (DMF), MIm, and PTSA were purchased from Adamas Reagent Co., Ltd. Shanghai, China. Terephthalic acid (H_2_BDC) anhydrous methanol, and benzaldehyde were obtained from Aladdin Industrial Corp. Shanghai, China.

### 2.2. Synthesis, Characterization, and Calculation of DESs

All DESs were prepared by heating appropriate mixture of MIm and PTSA with specific molar ratio [3MIm:PTSA] and [MIm:2PTSA] at 50 °C with continual stirring. Then mixture of MIm and PTSA with a homogeneous colorless solution by observation meant the formation of DES.

For the synthesis of UiO-66-DES, 1 mmol H_2_BDC was dissolved in 8.7 g of basic DES [3MIm:PTSA] and 1 mmol ZrOCl_2_·8H_2_O was dissolved in 9.4 g of acidic DES [MIm:2PTSA] at 50 °C. Then, the above two solutions were mixed and stirred at 50 °C for 2 h. The precipitate was collected by centrifugation, washed by de-ionized water three times, and dried under vacuum at 60 °C for 24 h.

As for UiO-66-DMF, 1 mmol H_2_BDC and 1 mmol ZrOCl_2_·8H_2_O were dissolved in 50 mL DMF. The mixed solution was then transferred to a 100 mL Teflon-lined autoclave and heated at 50 °C for desired time in a convection oven. The precipitate was collected by centrifugation, washed by de-ionized water three times, and dried under vacuum at 60 °C for 24 h. The product was then weighed to determine the yield of UiO-66, which was percent yield of experimental UiO-66 yield to the theoretical UiO-66 yield based on the calculation of initial reagents at a given reaction time.

The interaction between involved molecules and molecules or ions was calculated by a conductor-like screening model for real solvents (COSMO-RSs). The interaction energies and σ-potentials were calculated based on COSMOtherm 2022 (Version 22.0.0). The geometrical optimization of involved species was conducted by Gaussian 09 E.01 software package to form COSMO files, which were imported to COSMO-RS to calculate interaction energies.

Fourier transform infrared (FT-IR) spectra were recorded on a PerkinElmer Frontier™ spectrometer, range of 4000–400 cm^−1^ with 4 cm^−1^ step. Thermogravimetric and differential thermal analysis (TG-DTA) was performed on a PerkinElmer TGA-8000 with heating rate of 4 °C min^−1^ from 30 °C to 800 °C in flowing air of 200 mL·min^−1^. X-ray diffraction (XRD) data was collected on a Bruker D8 Advance diffractometer, using nickel-filtered CuKα radiation (λ = 1.54187 Å) in Bragg-Brentano geometry (U = 30 kV, I = 10 mA). Differential scanning calorimetry (DSC) was used to determine the freezing points of MIm-PTSA mixture (Pyris Diamond DSC, Waltham, MA, USA). The morphology was analyzed on a field emission scanning electron microscope (FE-SEM, JEOLJSM-6700F). Transmission electron microscope (TEM) image and particle size distribution of the sample was obtained on a JEOL JEM-1400 transmission electron microscope (TEM). The particle size distribution was evaluated more than 200 particles from the measurement of the TEM images. ^1^H NMR spectra were collected on a Bruker Advance III 400 NMR spectrometer, 400 MHz in DMSO. The pH of the aqueous MIm-PTSA mixture (0.01 mol·L^−1^ in water) was measured by Mettler Toledo FE-20 pH meter at 25 °C. The specific surface area of the samples was analyzed based on BET method using a Micromeritics ASAP2460 instrument, at 77 K.

### 2.3. Acetalization Experiments

The prepared UiO-66 catalysts were activated prior to catalytic test by drying the catalyst in vacuum at 300 °C for 3 h. 50 mg of catalyst, 3 mL methanol (74 mmol), 0.12 g of benzaldehydes (1.1 mmol) were added to a 10 mL sealed glass bottle. The reaction mixture was then stirred at 500 rpm at room temperature for 1 h. Reaction conversion was monitored by taking aliquots from the reaction mixture at different time intervals and analyzed by Agilent 7890 GC with a flame ionization detector and PEG-20 m column. The reaction and analysis of biomass feedstocks with methanol was performed under the same conditions above.

## 3. Results and Discussion

### 3.1. Fast Assembly of UiO-66

DESs with much low freezing points compared with each individual component, it is important to study the interaction of MIm and PTSA. Thus COSMO-RS, a statistical thermodynamics model based on quantum chemistry calculations, was employed to investigate the interaction effect between MIm and PTSA, as shown in Figure 1. Three regions are divided by hydrogen bond (HB) threshold of σ = ±0.0084 e/Å^2^. A molecule with σ-potential in the regions of σ < −0.0084, −0.0084 < σ < 0.0084, and σ > 0.0084 e/Å^2^ indicates its HBD, nonpolar, and HBA abilities, respectively. HBA and HBD abilities are the ability to form hydrogen bond involving electrostatic attraction, where a hydrogen is covalently bound to a more electronegative donor atom or group, namely HBD. On the other hand, the hydrogen acceptor is an electronegative atom of an adjacent molecule (HBA), containing a lone pair involved in the hydrogen bond. The deeper the red or blue color is, the stronger the HBA or HBD ability is. σ-Profile of MIm is broadly distributed in both polar regions, suggesting its strong HBA and HBD abilities. For PTSA, it also has a strong HBD ability, while the only peak in the HBA region and close to the non-polar region reflects its weak HBA ability. The analysis can also be confirmed by another important descriptor, σ-surface. As a result, MIm and PTSA are supposed to have a strong affinity each other through hydrogen bonds, and thus form DES through the decrease in freezing point.

Besides the interaction between MIm and PTSA, the interaction energy between PTSA and MIm cation or PTS anion is considered and displayed in Figure 2a. The HB interaction between PTSA and PTS cation is dominant among three major interaction energies including misfit, HB and van der Waals (vdW) interaction energies. The higher HB interaction energy between PTSA and PTS anion indicating the stronger HB ability significantly decreases freezing point of [MIm:2PTSA].

As for the interaction between MIm and MIm cation or PTS anion (Figure 2b), the HB interaction energy between MIm and MIm cation is close to that between MIm and PTS anion, which suggests that MIm can simultaneously interact with MIm cation and PTS anion. In addition, the vdW interaction and misfit energies between MIm and PTS anion are higher than those between MIm and MIm cation. Taken all in consideration, the PTS anion possesses strong interaction with MIm and PTSA, and it requires excess MIm to form DES [3MIm:PTSA] by the conjunct HB interaction between MIm and MIm cation as well.

The formation process of UiO-66-DES within acid-base tunable DESs formed from MIm and PTSA is described in Figure 1. The σ-surfaces of MIm, PTSA, [3MIm:PTSA], [MIm:2PTSA] in the scheme represent the actual molecular arrangement. Specifically, strong Brønsted acidic DES [MIm:2PTSA] (mole ratio) with pH = 1.8 is obtained, while weak basic DES [3MIm:PTSA] with pH = 7.4 is acquired in Appendix A. The freezing points of [MIm:2PTSA] and [3MIm:PTSA] are 16.0 °C and 39.0 °C, respectively, which are much lower than that of the pure individual components (142 °C for MIm and 103 °C for PTSA). The tuned acid-base properties have also been demonstrated in our previous work for imidazole-PTSA based DESs [36,37].

Herein, the weak basic [3MIm:PTSA] can completely dissolve the ligand (H_2_BDC) of UiO-66, and the strong acidic [MIm/2PTSA] can dissolve metal precursor ZrOCl_2_. The precursors were separately dissolved in acidic and basic DES solutions, and the mixture was just stirred under mild temperature (50 °C) for 2 h to produce UiO-66. Therefore, the tuned acid-based DESs can accelerate the formation of zirconium based MOF and high yield of MOF is obtained in a short reaction time in comparison with conventional solvothermal method [38,39].

UiO-66 with a formula Zr_6_O_4_(OH)_4_(BDC)_6_ has gained attractive applications in various fields owing to its superior thermal and chemical stability. Here the UiO-66 was synthesized in acid-base tunable DES based on MIm and PTSA. As the reaction time is shortened to 2 h, the relative intensities of the X-ray diffraction (XRD) peak of the solid product prepared in DES agree well with those of the reported UiO-66 and simulated one in Figure 3 [38,39]. In the contrast, the conventional solvothermal synthesis demands long reaction time to get good crystallinity of the MOF. When the UiO-66 was prepared in DMF with prolonged solvothermal time from 24 to 144 h, the intensities of the diffraction peaks gradually increase, which imply the UiO-66 synthesized in DMF requires long reaction time.

Generally, UiO-66 has a cubic structure with Fm-3m space group based on Zr oxo-clusters and BDC ligands [38]. The UiO-66 synthesized in DES (UiO-66-DES) has a unit cell of 20.810 Å and UiO-66-DMF synthesized in DMF with reaction time of 144 h has a unit cell of 20.717 Å, which are similar to the reported UiO-66. Average crystal sizes are also obtained according to Scherrer equation by measuring (111) diffraction peak at 2θ of about 7.3° in the XRD patterns. The UiO-66-DES and UiO-66-DMF have average crystal sizes of 64.2 and 58.5 nm, respectively. On basis of Ostwald ripening, a longer reaction time can efficiently improve crystallinity and yield of product, resulting in larger particle sizes [40]. Herein, UiO-66-DES has better crystallinity with larger crystal sizes within a shorter reaction time of 2 h, which indicates that acid-base tunable DES can accelerate the formation of the UiO-66.

Obviously, tuned acidic and basic DESs as solvents of metal precursor and organic ligand, respectively, significantly enhance the formation rate of UiO-66 compared with conventional organic solvent DMF. The yield of UiO-66-DES achieves the maximum of 87.5% after 2 h reaction time and is far higher than that of UiO-66-DMF with only 60.0% at 100 h (Appendix A). The results demonstrate that UiO-66-DES synthesized in DES exhibits high crystallinity and obtains high yield in a short reaction time. Furthermore, functionalized UiO-66 (UiO-66-NH_2_) and UiO-67 with extended phenyl linker (biphenyl-4,4′-dicarboxylate) can also be synthesized in the acid-base tunable DES with reaction time of 2 and 4 h, respectively, which are in well accord with previous works (Appendix A) [7,38].

### 3.2. Characterization of UiO-66

The fast formation of UiO-66-DES was further confirmed by FT-IR and Figure 4 shows FT-IR spectra of the two synthesized UiO-66. The bands at 1581 cm^−1^ and 1400 cm^−1^ can be attributed to ν(OCO) asymmetric and symmetric stretching of coordination modes, respectively [38]. The small band at 1506 cm^−1^ is due to C=C vibration of benzene ring in H_2_BDC ligands. Notably, the disappeared band at 1640–1670 cm^−1^ clearly shows the absence of DMF in the framework of UiO-66-DES, while DMF can be observed in UiO-66-DMF. Additionally, the band at around 1106 cm^−1^ presents the stretching vibration of Zr-O single bond in the framework. At lower frequencies, the OH and C-H bending modes are mixed with Zr-O vibration bands at 746 and 668 cm^−1^. The received information on H_2_BDC ligands, coordination modes, and Zr-O vibration by FT-IR analysis demonstrates that the UiO-66 framework can be easily and quickly synthesized in acid-based tunable DES.

The porosity of UiO-66-DES was studied through N_2_ adsorption/desorption analysis, which describes a Type I isotherm in nature with a narrow H2 hysteresis loop based on IUPAC classifications (Figure 5), the high crystallinity and the packing of crystal particles with micropores and mesopores (Figure 5, inset). The Brunauer-Emmett-Teller (BET) surface area of UiO-66-DES is 1146 m^2^·g^−1^, similar to previously reported results (1187 m^2^·g^−1^) via solvothermal process [38].

FE-SEM can be used to observe the morphology of the performed sample and the UiO-66-DES exhibits nanoparticle aggregates (Figure 6a), indicating the formation of small nanocrystals within a low reaction temperature and short reaction time of 2 h under the assistance of DES, rather than common octahedral shapes [41]. The TEM images (Figure 6b,c) show that the prepared UiO-66-DES is composed of small MOF nanoparticles, which also evidences the formation of mesopores. The UiO-66-DES nanoparticles have a size distribution of about 65 ± 25 nm (Figure 6d), which agree well with the average crystal size (64.2 nm) calculated from the Scherrer equation through the (111) diffraction peak of the XRD pattern (Figure 3).

It is reported that morphologies and crystal sizes of prepared UiO-66 strongly depend on precursors, additives, and synthesis methods [42]. Especially, the use of ZrOCl_2_·8H_2_O as the precursor can lead to aggregation of small MOF nanoparticles, which has been confirmed by very weak XRD patterns [43]. In this work, ZrOCl_2_·8H_2_O acted as precursor results in high crystallinity and small sizes of the UiO-66-DES (Figure 3 and Figure 6d). Moreover, the formation rate of the UiO-66-DES is significantly accelerated by the addition of the tuned acid-base DESs. PTSA in acidic [MIm:2PTSA] acts as a modulator to have a competition with H_2_BDC ligand for coordination sites on Zr-oxo clusters. MIm in basic [3MIm:PTSA] plays as a base to remove the proton of the carboxylic acid ligands, thus accelerating crystal nucleation and growth [44]. The DES can be selectively attached to the surface of the MOF, stabilizing the UiO-66 and restricting their growth, and thus small particle sizes are obtained.

Furthermore, the thermal and mass loss behaviors (TG-DTA) curved in Figure 7 demonstrate the UiO-66-DES is crystalline rather than amorphous structure. The reason is that the solvent loss and endothermic peak temperature of amorphous UiO-66 are far gentler and lower than those of crystalline one [45]. In general, crystalline UiO-66 has a distinct endothermic peak higher than 500 °C, while amorphous one has one strong endothermic peak lower than 300 °C and another wide endothermic peak from 400 °C. The strong endothermic peak at low temperature can be relevant to the composition transformation of high-density phase of amorphous UiO-66. Moreover, there is an evident endothermic peak at about 350 °C if excess linkers, unbound H_2_BDC molecules are resided in the pores of the material [39,45]. In this work, the UiO-66-DES has a distinct endothermic peak above 500 °C, demonstrating the crystalline structure in accord with XRD results. The curve shows two separate steps, which are attributed to the loss of residual solvent during heating to 300 °C and the subsequent decomposition of the linkers above 450 °C. The residual product at the end of TGA analysis is crystalline ZrO_2_ [39].

### 3.3. Formation Process of UiO-66

After verifying the significant effect of acid-base tunable DESs on the assembly of zirconium-based MOF (UiO-66-DES), the formation process was considered and revealed. The ^1^H NMR spectra of acidic DES, the addition of ZrOCl_2_, and the mixture after centrifugation, as shown in Figure 8. For the acidic DES, it is well known that the sulfonic acid group in PTSA as an electron-pair donor can form intermolecular hydrogen bond with imidazolium ring of MIm, resulting in down-field chemical shift (δ) at 13.94 ppm [46]. The PTSA of the DES may have similar effect to acetic acid, which can modulate linker H_2_BDC at the coordination sites of zirconium ion [27,41].

When ZrOCl_2_ was added into acidic DES [MIm:2PTSA], the typical peak of the H-N(1) atom in MIm at 6.16 ppm shifts to 5.59 ppm indicative of the formation of hydrogen bond between the acidic DES and ZrOCl_2_, which assists the coordination of ZrOCl_2_ to form polymeric hydroxide [27]. After the zirconium hydroxide nucleation by centrifugation of mixed ZrOCl_2_ and DES, the δ hydrogen of the H-N(1) atom in MIm is recovered. These results further clarify the coordination between ZrOCl_2_ and acidic DES, which has been supported by the coordination of ZrOCl_2_ and carboxylic acids (CH_3_COOH and HCOOH) [27].

The hydrogen bond effect is confirmed by FTIR (Appendix A), where absorption bands are similar except for different intensities for [3MIm:PTSA], [MIm:2PTSA], and [4MIm:3PTSA]. For PTSA and DESs, the peaks at 1004, 1031, and 1118 cm^−1^ are ascribed to –S=O groups. The stretching vibration of –S-OH at 863 cm^−1^ in pure PTSA, shifting to 916 cm^−1^ in [4MIm:3PTSA] and [MIm:2PTSA], is due to the formed hydrogen bond between –SO_3_H groups of PTSA and N-H groups of MIm [47]. In addition, the appearance of new peaks at 1960 cm^−1^ in [3MIm:PTSA], 2212 cm^−1^ in [MIm:PTSA], and 1932 cm^−1^ in [4MIm:PTSA] also demonstrates the N-H···O hydrogen bond of the DESs [48]. Furthermore, the enhanced intensities between 2500 and 3200 cm^−1^ show the strong interaction between and MIm and PTSA. Among them, the peak at 2919 cm^−1^ is attributable to the O-H···O hydrogen bond in DES [49].

Besides the ^1^H NMR spectra, FT-IR spectra in Figure 9 also support the coordination. The –OH groups of acidic DES at about 3435 cm^−1^ weakens and redshifts to 3482 cm^−1^ after the addition of ZrOCl_2_ into the DES, which indicates a weakened hydrogen bond induced from –SO_3_H groups in DES owing to coordination between ZrOCl_2_ and PTSA of DES. Moreover, the appearance of new should peaks at 897 and 1106 cm^−1^ beside the stretching vibration (863 cm^−1^) of –S-OH attributing to –Zr-OH and Zr-O vibrations also assists the coordination of ZrOCl_2_ and acidic DES [50].

The aforementioned analysis shows that PTSA in acidic DES and ZrOCl_2_ preliminarily forms Zr(IV) oxo organic acid framework. With the addition of mixed H_2_BDC linker and basic DES, the strong hydrogen bond effect between PTSA and MIm can accelerate the exchange of H_2_BDC and PTSA in Zr(IV) oxo organic acid framework [51]. This process has also been demonstrated by the acceleration of CH_3_COOH and HCOOH in ionic liquid for the formation of metal organic framework [27,50,51].

Thus, the formation process of UiO-66 nanoparticles using tuned acidic and basic DESs is figured out (Figure 2). Firstly, the dissolved ZrOCl_2_·8H_2_O in acidic DES quickly produces polymeric hydroxide Zr_4_(OH)_12_ via hydrolysis and dehydration (Figure 2I) [27]. The Zr_4_(OH)_12_ combing with the modulator PTSA in acidic DES forms Zr oxo organic acid coordination through hydrogen bonding confirmed by ^1^H NMR spectra in Figure 8 (Figure 2II). Then, the fast linker exchange between PTSA in acidic DES and H_2_BDC linker in basic DES occurs at the zirconium sites of the coordination [52].

The existed strong interaction (hydrogen bond) between PTSA and MIm greatly accelerates the linker exchange rate under mixed weak basic environment. In other words, MIm can promote the coordination between the linkers and zirconium oxo organic acid framework through its strong interaction with PTSA and increase the deprotonation rate of H_2_BDC linker. Consequently, the reaction between negatively charged BDC linker and zirconyl organic cation is facilitated, and the MOF cluster is produced rapidly in the weak basic DES at mild temperature (Figure 2III). Finally, the MOF clusters bridged by carboxylate groups from BDC linker leads to formation of the UiO-66-DES.

### 3.4. Benzaldehyde Acetalization with Methanol

The UiO-66-DES nanoparticles should have a large number of open zirconium sites, which may be highly active for some Lewis acid catalyzed reactions [53,54,55]. Acetalization is a typical synthetic protection for carbonyl group in ketones and aldehydes, which can be catalyzed by Lewis acidity [53]. Here the UiO-66-DES with open Lewis acid sites were used to evaluate benzaldehyde acetalization with methanol at room temperature (Table 1). In the absence of a catalyst, the reaction gives a 2% conversion (Table 1, entry 1), demonstrating weak proton acid catalyzed reaction with mixture acidity (pH 5.7). ZrOCl_2_, ZrCl_4_, and conventional ZnCl_2_ as homogeneous Lewis acid proton carrier show higher conversions (Table 1, entries 5–7). Additionally, the free ligand H_2_BDC with Brønsted protons is able to convert 13% of benzaldehyde, lower than those of Lewis acid catalysts demonstrating the Lewis acid catalyzed process (Table 1, entry 8).

Notably, UiO-66-DES achieves a conversion of 94%, far higher than those of homogeneous catalysts. For comparison, UiO-66-DMF is also evaluated and affords a competitive conversion (93%). Possessing the same linker H_2_BDC, Al and Cr based MOFs, Al_2_(BDC)_3_ and Cr(BDC), exhibit mild catalytic activities for the acetalization process (Table 1, entries 9, 10) [56,57]. In addition, Cu_3_(BTC)_2_ and Fe(BTC) with different linker (BTC=1,3,5-benzenetricarboxylate) also have been compared and show high catalytic activity within a long reaction time of 24 h (Table 1, entries 11, 12) [56]. These Lewis acid-based MOFs with open Lewis acid sites with various metal ions provide appropriate catalytic activities, however all lower than UiO-66-DES. Therefore, UiO-66-DES nanoparticles with small particles sizes possessing a large amount of open zirconium catalytic sites exhibits high catalytic activity for the acetalization of benzaldehyde with methanol.

Besides benzaldehyde, furfural and 5-hydroxymethylfurfural, as very important biomass feedstocks, were tested to further verify the Lewis acid catalyzed acetalization reactions (Appendix A). 5-Hydroxymethylfurfural can be acetalized with 93% conversion and the selected aldehydes are acetalized to their corresponding acetals with high aldehyde conversion. The high catalytic efficiency of UiO-66-DES is originated from open zirconium Lewis acid sites and small nanosizes, facilitating the mass transfer and accessibility of substrates to active sites.

## 4. Conclusions

In summary, we offered a rapid synthetic route to produce UiO-66 nanoparticles using tuned acidic and basic DESs, which was formed from the same hydrogen bond donor and hydrogen bond acceptor, as solvent without any aggressive additives under facile reaction conditions. The formation mechanism of the UiO-66 in the acid-base tunable DES has been systemically investigated and proposed. Furthermore, the prepared UiO-66 with small nanosized diameters and a large amount of open zirconium Lewis acid sites facilitated the mass transfer and accessibility of substrates to active sites. Thus, it exhibited improved catalytic activity for benzaldehyde, furfural, and 5-hydroxymethylfurfural acetalizations with methanol. Therefore, the fast synthesized method has the potential to prepare versatile MOFs with various functions and pore diameters and metal oxides in acid-based tunable DESs under mild conditions for other attractive applications, especially in separation, gas adsorption, and storage applications.

## Data Availability

Not applicable.

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
