# Peer review of "Fast Assembly of Metal Organic Framework UiO-66 in Acid-Base Tunable Deep Eutectic Solvent for the Acetalization of Benzaldehyde and Methanol"

_molecules, 2022, doi:10.3390/molecules27217246_

Round 1

Reviewer 1 Report

In this paper, Chen et al reported a successful alternative UiO-66 prep using DES. The materials were characterized and the formation mechanism was discussed. Further, the UiO-66-DES exhibited comparable catalytic performance with its solvothermally prepared counterpart. The manuscript is well-composed, it is recommended for publication after addressing the following issues.

First of all, what’s the yield for UiO-66, under this new condition, as a result of lowered temperature and shortened rxn time?

Next, on page 6, clearly there is a typo “H2 hysteresis loop” when describing the N2 sorption isotherms.

On page 6-7, it is understandable that the UiO-66 formed in DES at low T and short rxn time are small, but one TEM image does not suffice. Do the authors have DLS results or other images accompanied by statistical diagram?

Last but not least, on page 7, the authors mentioned that “the curve shows two separate steps, which are attributed to the loss of guest molecules, the amorphization of the UiO-66-DES during heating to 300 C”. If the MOFs are losing guest molecules as gases and the crystals are slowly turning into amorphous solids, wouldn’t there be significant peaks on the DTA profile? Moreover, in the Experimental section, for catalysis, the MOFs were pre-heated at 300 C, which is self-conflicted to the TGA results description. PXRD after thermal treatment (300 C) is necessary here to reveal if the MOFs are still MOFs at this temperature, if they are, the “amorphization” is incorrect; if they are not, the preparation protocol of catalysts needs to be changed if the authors still want to use (and report) the MOFs rather than partially decomposed products (some may have started to turn into oxides and stuff) as catalysts.

Reviewer 2 Report

The manuscript is well written. It reports several findings that are insightful in the field of green solvents application (DESs) and solid adsorbent development (MOFs).

Please consider the following comments:

- Section 3.1: Other than the analysis of hydrogen bonding between different species, it is also suggested to include discussion on the self-affinity, i.e. affinity of MIm with other MIm molecules; as well as the affinity of PTSA within other PTSA molecules.

- In Scheme 1, does the sigma surface of [3MIm:PTSA] and [MIm:2PTSA] represent the actual molecular arrangement?

- In page 6, "The BET surface area of UiO-66-DES is 1146 m2·g-1, similar to previously reported via solvothermal process" - it is suggested to mention the exact value in the solvothermal process.

- In page 6, the authors also cited Figure 5b, but Figure 5 has no sub-figures.

- In section 3.3, authors wrote "The 1H NMR spectrum of acidic DES shows that DES has a distinguish difference with that of pure PTSA (Figure 7).", but Figure 7 shows the 1H NMR spectra of DES vs MOF-DES (not pure PTSA). Please check this part.

Reviewer 3 Report

The article written by Lifang Chen and co-authors: ‘Fast Assembly of Metal Organic Framework UiO-66 in Acid-Base Tunable Deep Eutectic Solvent for the Acetalization of Benzaldehyde and Methanol’ presents a new strategy for the synthesis of zirconium organic frameworks (Zr(IV)-MOFs) under mild chemical reaction conditions, and in a relatively short time. The introduction is sufficient to bring in the subject of research. The chemistry and methodology could be extended by more details, e.g. the name and country of the company where the chemicals were pursued, the level of grade purity, etc. The results and discussion are accurate and well presented. But the conclusions could be extended about the possible applications of received UiO-66 nanoparticles. Therefore, I have a few suggestions to improve the quality of the manuscript, which have to be corrected, and are listed below.

1.      Why are the lines in the article not numbered? Their absence makes reviewing difficult.

2.      The chapter subsections from 2.2 to 2.5 should be in one section but in separate paragraphs.

3.      Did you examine the level of purity final product? What was the purity and yield at each stage of synthesis?

4.      The list of abbreviation need to be added at the end of the article.

5.      Do you know how the XRD patterns of UiO-66-DMF will look after a more extended synthesis than 2h? How do you think a longer reaction time could improve crystallinity and yield?

6.      At FT-IR spectra (Figure 8, Figure S4) and 1H NMR spectra (Figure 7) please mark the position of the main peaks which are mentioned in the text.

7.      At page 9, the sentence: ‘The Zr4(OH)12 combing with the modulator PTSA in acidic DES forms Zr oxo organic acid coordination through hydrogen bonding confirmed by 1H NMR spectra in Figure 6a (Scheme 2II).’, something went wrong because in Figure 6 is TG-DTA curves, and there is no part a, b. Please correct this mistake.

8.      It would be good if some of the Figures were bigger. Especially those in supplementary information and Scheme 2.

9.      In Table 1 columns time and conversion %: decimal numbers should not be written with a semicolon ‘;’, please correct it.

10.  The summary part should include a broader description of the applications and uses of the obtained Zr(IV)-MOFs.

11.  The authors contribution statement has to be added.

12.  In the article, there are editorial bugs and punctuation errors that have to be corrected before further evaluation. My advice is to make an editorial revision of whole the article.

Reviewer 4 Report

The manuscript shows interesting results on synthetic approach to preparation of well-known MOF Zr-UiO66. However, before accepted the manuscript requires following serious corrections:

1. Figure 1 shows σ-Profiles of 2-methyl imidazole and p-toluenesulfonic acid. These data look as results of theoretic calculations, nevertheless no details or references are described in manuscript.

2. Page 4, 1st paragraph: ‘The formation process of DES between MIm and PTSA was described in Scheme 1.

Specifically, strong Brønsted acidic DES [MIm:2PTSA] (mole ratio) with pH = 1.8 is obtained, while weak basic DES [3MIm:PTSA] with pH = 7.4 is acquired in Table S1, Supporting Information. The freezing points of [MIm:2PTSA] and [3MIm:PTSA] are 16.0 oC and 39.0 oC, respectively, which are much lower than that of pure individual component

(142 oC for MIm and 103 oC for PTSA).’ Experiments on determination of melting point of two-phase mixtures are not described. Moreover, from the manuscript (Table S1) it is not clear if the composition 3MIm:PTSA really corresponds to eutectic composition, since the 4MIm:PTSA shows lower melting point. It is also possible the formation of individual complex compound, therefore the phase analysis of solid phase is required. Please, provide the phase diagram of the certain solvents system if it is known.

3. Page 5, 1st paragraph, ‘Moreover, the (111) peak intensity of the UiO-66 synthesized in DES (UiO-66-DES) is narrower and higher than that of the UiO-66 synthesized in DMF with reaction time of 144 h (UiO-66-DMF), indicating UiO-66-DES has be?er crystallinity with shorter reaction time of 2 h.’ The absolute intensity on XRD pattern depends on many factors, among them are the amount of sample material which varies from sample to sample due to different packing density. Therefore, this criterion can not be used to estimate the crystallinity. The crystallinity can be estimated from the peaks FWHM, at lease by Scherrer equation. Please, give the results of such analysis.

4. Page 3, section 3.1., Please give the definition of ‘HBA and HBD abilities

Typos:

page 2, section 2.1: ‘N,N-dimethyllformamide

Round 2

Reviewer 1 Report

the authors have satisfactorily addressed all issues raised by this reviewer, hence it is now recommended for acceptance for publication.

Reviewer 3 Report

The article written by Lifang Chen and co-authors: ‘Fast Assembly of Metal Organic Framework UiO-66 in Acid-Base Tunable Deep Eutectic Solvent for the Acetalization of Benzaldehyde and Methanol’, which I reviewed earlier new version importance has gained in quality. All corrections were made, so  I recommend this manuscript to publish in Molecules. 

Reviewer 4 Report

Authors addressed all reviewer comments. The manuscript can be accepted in present form after minor texual corrections:
1. page 3, ' X-ray diffraction (XRD) data was collected on a Bruker D8 Advance diffractometer, using nickel-filtered CuKα radiation (λ = 1.54056 Å)' The average wavelength of CuKα radiation λ = 1.54187 Å.
2. page 11, '3.4. benzaldehyde acetalization with methanol' The title should start from capital letter.